# Comparison of Widely Targeted Metabolomics and Untargeted Metabolomics of Wild *Ophiocordyceps sinensis*

**DOI:** 10.3390/molecules27113645

**Published:** 2022-06-06

**Authors:** Jinna Zhou, Donghai Hou, Weiqiu Zou, Jinhu Wang, Run Luo, Mu Wang, Hong Yu

**Affiliations:** 1College of Science, Tibet University, Lhasa 850001, China; 18487320559@163.com (J.Z.); xiaoyaozi5188@sina.com (J.W.); 2School of Life Sciences, Yunnan University, Kunming 650106, China; donghaihou@126.com (D.H.); zouweiq@mail.ynu.edu.cn (W.Z.); luorun0214@163.com (R.L.); 3Plant Science College, Tibet Agriculture & Animal Husbandry University, Nyingchi 851418, China; 4Yunnan Herbal Laboratory, College of Ecology and Environmental Sciences, Yunnan University, Kunming 650106, China

**Keywords:** metabolomics, *Ophiocordyceps sinensis*, widely targeted metabolomics, untargeted metabolomics, Fungi

## Abstract

The authors of this paper conducted a comparative metabolomic analysis of *Ophiocordyceps sinensis* (OS), providing the metabolic profiles of the stroma (OSBSz) and sclerotia (OSBSh) of OS by widely targeted metabolomics and untargeted metabolomics. The results showed that 778 and 1449 metabolites were identified by the widely targeted metabolomics and untargeted metabolomics approaches, respectively. The metabolites in OSBSz and OSBSh are significantly differentiated; 71 and 96 differentially expressed metabolites were identified by the widely targeted metabolomics and untargeted metabolomics approaches, respectively. This suggests that these 71 metabolites (riboflavine, tripdiolide, bromocriptine, lumichrome, tetrahymanol, citrostadienol, etc.) and 96 metabolites (sancycline, vignatic acid B, pirbuterol, rubrophen, epalrestat, etc.) are potential biomarkers. 4-Hydroxybenzaldehyde, arginine, and lumichrome were common differentially expressed metabolites. Using the widely targeted metabolomics approach, the key pathways identified that are involved in creating the differentiation between OSBSz and OSBSh may be nicotinate and nicotinamide metabolism, thiamine metabolism, riboflavin metabolism, glycine, serine, and threonine metabolism, and arginine biosynthesis. The differentially expressed metabolites identified using the untargeted metabolomics approach were mainly involved in arginine biosynthesis, terpenoid backbone biosynthesis, porphyrin and chlorophyll metabolism, and cysteine and methionine metabolism. The purpose of this research was to provide support for the assessment of the differences between the stroma and sclerotia, to furnish a material basis for the evaluation of the physical effects of OS, and to provide a reference for the selection of detection methods for the metabolomics of OS.

## 1. Introduction

*Ophiocordyceps sinensis* (Berk.) (OS) Sung et al., is a parasitic complex of the fungus *O. sinensis* [1], which is distributed in the Qinghai-Tibet Plateau and surrounding areas at an altitude of 3200–5300 m. Modern pharmacological studies have found that OS exerts different degrees of therapeutic effects on the cardiovascular system, respiratory system, nervous system, immune system, kidney, and liver [2,3,4,5,6]. In addition, it has antitumor and antioxidant activities [7,8]. The pharmacological action of OS is closely related to its polysaccharides, nucleosides, sterols, flavonoids, cyclic peptides, phenols, anthracenes, polyketones, and alkaloids [9,10]. Most research on the active components of OS is focused on adenosine and ergosterol. For example, Yang et al. found that there was a significant difference in adenosine content between the stroma and sclerotia of OS (OSBSz and OSBSh, respectively) [11]. A recent study showed that ergosterol is more abundant in hosts than in the stroma, especially on top of the host. Intensive research on the known active components of OS has been conducted, and potential markers have been elucidated. Zhong et al. found that the water extract of OS had significant anti-inflammatory effects against single stimulation by cigarette smoke extract (CSE), as well as infection by a compound influenza virus [12]. A UHPLC-Q-TOF-MS technique was used to extract a novel fatty acid—(2Z,4e)-deca-2,4-dienoic acid—from CSE, which attenuated the inflammatory response by reducing mRNA and protein levels in cells [13]. The same study also found that OS contains a water-insoluble polysaccharide, β-(1,3) glucan, which is not only a component of fungal cell walls, but it is also the core structure of immunologically active polysaccharides that stimulate immune receptors such as Dectin-1 to trigger innate immune responses; this suggests that there may a large number of unknown active ingredients in OS, which could play an important role in the prevention and treatment of diseases. However, the identification of potential markers is challenging, and there is a need for new methods to find accurate biomarkers.

In previous studies, chromatographic and spectroscopic techniques have been widely used in the study of OS, such as thin layer chromatography [14], high-performance liquid chromatography (HPLC) [15], gas chromatography [16,17], capillary electrophoresis chromatography [18], infrared spectroscopy [19], and nuclear magnetic resonance spectroscopy [20]. The use of these techniques has rapidly advanced the research of the chemical composition and quality evaluation of OS. HPLC is currently the most widely used technique in the research of OS. Detection and analysis are performed with different detectors depending on the nature of the analyte. Mass spectrometric (MS) detectors provide structural information and have high sensitivity. For example, cyclo-Ala-Leu-rhamnose and Phe-O-glucose were identified in OS by using HPLC-MS [21].

In addition to being used for quality assessment, HPLC-MS metabolomics is also becoming increasingly applied in discovery studies for potential biomarkers. It is well known that genomics and proteomics explore activities at the gene and protein levels, respectively, but in fact, many activities in cells occur at the metabolite level; for example, cellular signaling, energy transfer, and intercellular communication are regulated by metabolites [22,23]. Metabolomics is a static concept, which links differentially expressed metabolites to phenotypic changes through the qualitative and quantitative analysis of low-molecular-weight metabolites in different samples, which can identify metabolites that play important roles and preliminarily explore the causes of the changes. Such findings are regarded as a “bellwether” in the process of biomarker research for OS. Therefore, the choice of detection technique is crucial for OS research. A rigorous and simple detection technique can not only obtain credible results but also provide technical references for the following research. Metabolomics can not only be applied to explain the interaction mechanism between species, but also to explore new metabolites with biological activity [24]. Metabolomics is categorized as untargeted metabolomics or targeted metabolomics, depending on the aim of the study. Untargeted metabolomics is applied to detect all small molecule metabolites in an unbiased manner, whereas targeted metabolomics is applied to quantify only the targeted metabolites of interest. Widely targeted metabolomics is a novel technique that differs from existing metabolite detection methods because it integrates the advantages of the “generality” of untargeted metabolomics and the “accuracy” of targeted metabolomics. Furthermore, widely targeted metabolomics is a high-throughput, ultrasensitive technique with wide coverage that can be used for both qualitative and quantitative analysis.

At present, in the metabolomic research on OS, comparative research of untargeted metabolomics and widely targeted metabolomics is still lacking. Therefore, the purpose of the present study was to explore the metabolic profiles of the stroma and the sclerotia of OS using different detection techniques, and to identify important metabolites and key metabolic pathways by combining multivariate statistical analysis methods. This could provide technical ideas for future research, provide important references for the choice of detection technology, and also provide favorable support for the study of the medicinal value of OS.

## 2. Results

### 2.1. Metabolite Identification

Representative total ion chromatograms of OSBSz and OSBSh obtained by widely targeted metabolomics are shown in Figure 1a. As can be seen from the figure, all target compounds exhibited symmetrical chromatographic peaks, and chromatographic separation of individual target compounds was well achieved. From the base peak chromatogram obtained using untargeted metabolomics (Figure 1b), it can be seen that the samples had a good peak shape and large peak capacity. After peak deconvolution, alignment, and exclusion of ion features, 3491 ions were acquired by untargeted metabolomics. In total, the 778 and 1449 metabolites were tentatively identified in the chromatograms using widely targeted metabolomics and untargeted metabolomics, respectively, by comparing their fragment patterns with a mass spectral database. In terms of the types of metabolites, they were mainly amino acids, terpenoids, phenylpropanoids, alcohols, aromatics, phenols, nucleosides, flavonoids, and alkaloids, which is in line with the main types of fungal metabolites.

### 2.2. Principal Component Analysis

As the most widely used unsupervised pattern recognition method, principal component analysis (PCA) can intuitively demonstrate the overall distribution of samples. The R^2^ value obtained from PCA is regarded as an important indicator, with larger R^2^ values indicating better fitting of the model. As can be seen from the PCA score plots (Figure 2a,b), all samples were within 95% confidence intervals, whether using widely targeted metabolomics (R^2^X = 0.703) or untargeted metabolomics (R^2^X = 0.685). Three biological replicates were clustered, demonstrating good experimental reproducibility. OSBSz showed a separation trend from OSBSh, which indicates that the metabolites of OSBSz were significantly different from those of OSBSh. To obtain reliable and high-quality metabolomic data, quality control (QC) samples are usually used for quality control during detection. The smaller the difference between QC samples, the higher the stability of the method and the higher the quality of the data. The dense distribution of two QC samples in the PCA score chart shows that the data are reliable.

### 2.3. Partial Least Squares Discriminant Analysis

PCA may cause misjudgment in classification, and data must be further analyzed by supervised partial least squares discriminant analysis (PLS-DA). Compared with PCA, PLS-DA has a stronger ability to extract variation information between groups. From the PLS-DA score plots (Figure 2c,d), it can be seen that OSBSz and OSBSh performed very differently, data points of OSBSz and OSBSh were significantly distinguished, regardless of which detection means were applied, which was consistent with the PCA results. The model fitting parameter (R^2^X) of the widely targeted metabolomics and untargeted metabolomics approaches was 48.3% and 63.1%, respectively, and the model discrimination parameter (R^2^Y) was 89.1% and 99.1%, respectively, indicating that the model had a good fit. To check if the PLS-DA model was overfitted, 200 permutation tests were performed [24]. As shown in Figure 2e,f, the R^2^ and Q^2^ values in the original model were very close to 1, which proved that the established model conformed to the real situation of the sample data. This means that the PLS-DA model was not overfitted and the model was reliable.

### 2.4. Screening of Differentially Expressed Metabolites

As shown in Appendix A, the widely targeted metabolomics and untargeted metabolomics approaches were employed to screen out 70 and 96 differentially expressed metabolites, respectively. As shown in Figure 3a,b, the differentially expressed metabolites mainly included amino acids, nucleosides, alcohols, alkaloids, esters, and flavones. The differentially expressed metabolites identified by widely targeted metabolomics mainly included amino acid, nucleoside, and acid metabolites, such as threonine, proline, arginine, asparagine, gemcitabine, flavin adenine dinucleotide (FAD), and 5-Methyltetrahydrofolic acid. The differentially expressed metabolites identified by untargeted metabolomics mainly included tyrosine, arginine, DOPA, epalrestat, indospicine, and 2’-Deoxycytidine. Furthermore, metabolites such as 4-Hydroxybenzaldehyde, arginine, and lumichrome were identified by both techniques. These results constitute encouraging preliminary findings in favor of the use of metabolomics to explore potential biomarkers.

Comparative analysis between OSBSh and OSBSz using the widely targeted metabolomics approach showed that OSBSh contained more bromocriptine, lumichrome, pyridoxine, arginine, and 2-picolinic acid than OSBSz (Figure 4a). Perhaps these metabolites are important components of the sclerotia and closely related to the physiological activities of the host. For example, arginine participates in biochemical reactions such as ammonia detoxification and the immune system. Arginine is an important metabolite to maintain the survival of the host. 5-Methyltetrahydrofolic acid, nicotinic acid adenine dinucleotide, tetrahymanol, citrostadienol, melibiose, and salsolinol were more abundant in OSBSz. Using the untargeted metabolomics approach, the levels of sancycline, glycyrin, pirbuterol, neotame, rubrophen, geniposidic acid, arginine, methionylleucine, 4-Pyridoxic acid, and ethosuximide were higher in OSBSh. 3-Phenylpropanoic acid, phthalic acid, and epalrestat were more abundant in OSBSz (Figure 4b). These metabolites may be involved in the germination, maturation, and aging of the stroma.

### 2.5. Hierarchical Cluster Analysis

Hierarchical cluster analysis (HCA) of metabolites based on their characteristics can be used to classify the metabolites that have the same characteristics into one group, and to then find the variation of metabolites between OSBSz and OSBSh. HCA was applied to the top 20 differentially expressed metabolites. The results showed that using the widely targeted metabolomics approach (Figure 5a), nicotinic acid adenine dinucleotide, oxymorphone, and homoserine were highly expressed in OSBSz, whereas 5-aminovaleric acid, 6-aminocaproic acid, 2-picolinic acid, betaine aldehyde, riboflavin, tripdiolide, lumichrome, ornithine, proline, citrulline, arginine, asparagine, threonine, 4-Hydroxybenzaldehyde, and histidine were highly expressed in OSBSh. Among them, arginine was also highly expressed in OSBSh in the untargeted metabolomics, which was consistent with the top 20 differentially expressed metabolites chart. These results suggested that the differences between OSBSz and OSBSh may be regulated by these key substances.

### 2.6. Analysis of Differentially Expressed Metabolite Pathways and Enrichment Analysis

Pathway enrichment analysis revealed that the enriched pathways were mainly involved in metabolism and then biosynthesis. In total, 18 and 12 metabolic pathways were enriched, respectively (Figure 6a,b). As shown in the bubble chart, the abscissa where the bubble is located and the size of the bubble represent the influence value. The larger the bubble, the greater the importance of the pathway. The ordinate where the bubble is located and the color of the bubble represent the *p* value of the enrichment analysis. The redder the bubble, the closer the *p* value is to 0, hence the enrichment is more significant. It was found that the differentially expressed metabolites detected by widely targeted metabolomics were primarily involved in nicotinate and nicotinamide metabolism, thiamine metabolism, riboflavin metabolism, glycine, serine, and threonine metabolism, and arginine biosynthesis (Figure 6c). The differentially expressed metabolites detected by the untargeted metabolomics were mainly involved in arginine biosynthesis, terpenoid backbone biosynthesis, porphyrin and chlorophyll metabolism, and cysteine and methionine metabolism (Figure 6d). Specifically, these metabolic pathways may be the key to determine the differences between OSBSz and OSBSh, and important differentially expressed metabolites play a crucial role in the key pathways. The common metabolic pathways of the differentially expressed metabolites were screened by the two methods, including arginine biosynthesis, cysteine and methionine metabolism, arginine and proline metabolism, aminoacyl-tRNA biosynthesis, porphyrin and chlorophyll metabolism, and pyrimidine metabolism. Arginine is found among the differentially expressed metabolites of the two detection methods, and arginine is involved in normal activities of the body as an important amino acid.

## 3. Discussion

To the best of our knowledge, this is the first study to compare widely targeted metabolomics and untargeted metabolomics combined with multivariate data analysis approaches to explore the metabolite differences between OSBSz and OSBSh. The identification of metabolites has always been an important challenge in metabolomics research. At present, there is no metabolomic analysis platform that covers all metabolites without bias [25,26]. In this study, two detection methods brought different results. The number of metabolites identified varied greatly; 778 and 1449 metabolites were detected by widely targeted and untargeted metabolomics, respectively. There are two main reasons for this difference. First, there are differences in underlying qualitative principles between the two detection modalities. For the widely targeted metabolomics approach, the data acquisition principle is multiple reaction monitoring (MRM) technology based on triple quadrupole MS. The noise interference of MS signals can be significantly reduced under the dual mass screening of parent ions and daughter ions; their identification is more accurate, reproducible, and sensitive [27]. However, untargeted metabolomics relies on the parent ion’s signal intensity when selecting it for fragmentation and secondary scanning, which is performed on parent ions that meet a threshold range according to a set signal threshold of parent ions; for this reason, metabolites were detected in higher numbers by untargeted metabolomics than by widely targeted metabolomics. Second, the extraction of metabolites is a crucial part of metabolomics studies, which directly affects the range of detectable metabolites as well as the number of metabolites extracted [28,29]. In the present study, the extraction solutions for the widely targeted and untargeted metabolomics were methanol:H_2_O (3:1) and methanol:acetonitrile:H_2_O (2:2:1), respectively. The addition of acetonitrile, included because of its better separation, symmetrical peak shape, and stable baseline, enabled the solution to sufficiently extract the metabolites of OS. The extraction of metabolites is a key part of the experiment, and effective extraction is the premise for the successful detection of metabolites. Therefore, accurate identification of metabolites is necessary in the metabolomics study of OS.

Scientific acquisition of differentially expressed metabolites is another challenge in metabolomics [30]. During multivariate data analysis, since unsupervised PCA causes false positives when classifying, analysis by supervised PLS-DA is necessary to visualize the overall distribution of the samples. Irrespective of the detection method used, significant differences were found between OSBSz and OSBSh. Metabolites with VIP > 1 and *p* < 0.05 were considered as differentially expressed metabolites in this study [31,32]. In previous studies, some important differential metabolites showed significant biological activity. Epalrestat is used to treat diabetes, kidney disease, heart disease, and retinopathy and can relieve diabetic neuropathic pain [33,34]. Flavin adenine dinucleotide (FAD) is an essential component of the body’s intracellular multifunctional oxidase system, involved in intracellular redox and electron transport systems in mitochondria, and associated with in vivo metabolism of sugars, fats, proteins, etc. Moreover, FAD can be used to lower blood pressure [35]. DOPA is an effective drug for the prevention and treatment of Parkinson’s disease, carbon monoxide poisoning, and other conditions [36]. Therefore, the differentially expressed metabolites provide a material basis for revealing the pharmacodynamics of OS and may be potential biomarkers for OS. Both assays focused on 4-Hydroxybenzaldehyde, arginine, and lumichrome. 4-Hydroxybenzaldehyde is one of the main active components of *Gastrodia*
*elata*, which exerts protective effects against ischemic stroke. Furthermore, it inhibits thrombus formation, protects the blood–brain barrier, and relaxes vascular smooth muscle. It is expected to be developed into a new drug for the prevention and treatment of ischemic stroke [37]. It may play an important role in the treatment of cardiovascular and cerebrovascular diseases. Arginine is a semi-essential amino acid, which is involved in protein biosynthesis and host immune reactions. Clinically, arginine supplementation is used to treat some cardiovascular diseases, such as hypertension and coronary heart disease [38]. Lumichome is one of the photodegradants of vitamin B_2_. It participates in biological oxidation in vivo, is related to energy metabolism and carbohydrate, protein, nucleic acid, and fat metabolism, and also shows antitumor activity [39]. Based on these observations, it can be speculated that 4-Hydroxybenzaldehyde, arginine, and lumichrome are extremely important in the growth process of OS, and they may be important effective components for the prevention and treatment of diseases by OS.

The purpose of enrichment analysis is finding biological pathways that play a key role in a biological process, and revealing and understanding the basic molecular mechanisms of biological processes. In this study, we found that the key pathways in which the important differentially expressed metabolites of OS are mainly involved were arginine biosynthesis, cysteine and methionine metabolism, arginine and proline metabolism, aminoacyl-tRNA biosynthesis, porphyrin and chlorophyll metabolism, and pyrimidine metabolism. The microbial/ plant arginine biosynthesis pathway offers the potential capacity of antimicrobial and biocidal benefits [40]. Gut microbiota played a critical role in promoting the host resistance to low-temperature stress in *Bactrocera dorsalis* by regulating its arginine and proline metabolism pathway [41]. The aminoacyl-tRNA biosynthesis pathway is important for protein synthesis and is most strongly associated with the detoxification of ammonia [42]. This suggests that the low temperature resistance, biological control and nutrient absorption of OS is related to key pathways. The shortcomings of both methods could only be relatively quantified, and due to the small number of replicates, more common components were not obtained in the analysis of differentially expressed metabolites. It was suggested that the database of fungal resources should be expanded, refined, and specialized so that there is symmetry between the two methods that were compared. In future metabolome studies of OS, detection methods should be selected according to the purpose of the study. To obtain oridonin metabolic fingerprints, to obtain more comprehensive metabolite information, or to rapidly classify the samples, untargeted metabolomics is perhaps the best choice. Widely targeted metabolomics can be used for a focused analysis of the metabolites associated with a certain class of compounds for a specific metabolic pathway, for oridonin quality assessment, and for the identification of metabolites. If untargeted metabolomics was first used for preliminary metabolite targeting, further targeted exploration with widely targeted metabolomics may maximize the information obtained by metabolomics.

In summary, the present study provided the metabolic profiles of different parts of OS as well as references for quality assessment, further research of the chemical composition of OS, and phenotype analysis. Moreover, by multivariate statistical analysis, potential biomarkers, and key metabolic pathways were highlighted, which provide support for the study of the pharmacological effects of OS. Importantly, the contrasting metabolome results of the present study provide a basis for method selection in future metabolome studies.

## 4. Materials and Methods

### 4.1. Materials

Wild OS from Baima Snow Mountains were collected in June 2021. The bacterial membrane was first washed with distilled water, and OS was cut from the base of the stroma into two parts, the stroma and sclerotia. The stroma was labeled as OSBSz-x and the sclerotia was labeled as OSBSh-x. Three biological replicates were used for each group.

Instruments for widely targeted metabolomics include ultrahigh-performance liquid chromatograph (Sciex, Framingham, MA, USA), a high-sensitivity mass spectrometer (Sciex, Framingham, MA, USA), and a centrifuge (Thermo Scientific, Waltham, MA, USA), and a water purification system (Merck Millipore, Burlington, MA, USA). Instruments for untargeted metabolomics include ultrahigh-performance liquid chromatography (Waters, Milford, MA, USA), a high-resolution mass spectrometer (Thermo Scientific, Waltham, MA, USA), a low-temperature high-speed centrifuge (Eppendorf, Hamburg, Germany), a vortex (Qilin Bell Instrument Manufacturing Co., Ltd., Haimen, China), a water purification system (Millipore, Burlington, MA, USA), and a refrigerated vacuum concentrator (Gene company limited, Hong Kong, China).

### 4.2. Metabolite Extraction

In the widely targeted metabolomics approach, freeze-dried samples were crushed with a mixer mill for 1 min at 60 Hz. Then a 100-mg aliquot of each sample was transferred to an Eppendorf tube. After the addition of 1.5 mL of extraction solution (methanol:water, 3:1, precooled at −40 °C, containing internal standard), extraction was performed overnight at 4 °C on a shaker. Then samples were centrifuged at 12,000 rpm for 15 min at 4 °C. The supernatant was carefully filtered through a 0.22 μm microporous membrane and transferred into a fresh 2 mL glass vial. From each sample, 40 μL was taken, and these aliquots were pooled as QC samples and stored at −80 °C until UHPLC-MS analysis [43,44].

In the untargeted metabolomics approach, after thawing the samples slowly at 4 °C, 25 mg was put into a 1.5-mL Eppendorf tube. Next, 800 μL of the extraction solution (methanol:acetonitrile:water, 2:2:1, precooled at −20 °C) was added, and 10 μL of the internal standard and two small steel balls were added. Samples were placed in a tissue grinder (50 Hz, 5 min). After ultrasonic treatment in a water bath at 4 °C for 10 min, samples were stored at −20 °C for 1 h. Centrifugation was performed at 25,000 rpm at 4 °C for 15 min. After centrifugation, 600 μL of the supernatant was removed and drained by a vacuum concentrator. Next, a 200 μL complex solution (methanol:H_2_O, 1:9) was added for redissolution, followed by vortex shock for 1 min, ultrasonic treatment in a water bath at 4 °C for 10 min, and centrifugation at 25,000 rpm for 15 min at 4 °C. The resulting supernatant was used for further analysis [45,46].

### 4.3. UHPLC-MS Conditions

In the widely targeted metabolomics experiments, UHPLC separation was carried out using an ExionLC system. Mobile phase A was 0.1% formic acid in water and mobile phase B was acetonitrile. The column temperature was set at 40 °C. The auto-sampler temperature was set at 4 °C and the injection volume was 2 μL. The gradient for chromatographic analysis was as follows: 0–0.5 min, 2% B; 0.5–10 min, 50% B; 10–13 min, 95% B; 13–15 min, 2% B.

A Sciex QTrap 6500+ was applied for assay development. Typical ion source parameters were: ion spray voltage: +5500/−4500 V; curtain gas: 35 psi; temperature: 400 °C; ion source gas 1 pressure: 60 psi; ion source gas 2 pressure: 60 psi; DP: ±100 V.

In the untargeted metabolomics experiments, the chromatographic column used was BEH C_18_ (1.7 μm, 2.1 × 100 mm, Waters, Milford, MA, USA). The positive ion mode mobile phase was an aqueous solution containing 0.1% formic acid (liquid A) and methanol containing 0.1% formic acid (liquid B), and the negative ion mode mobile phase was an aqueous solution containing 10 mM ammonia formic acid (liquid A) and 95% methanol containing 10 mM ammonia formic acid (liquid B). The column temperature was set at 45 °C. The flow rate was 0.35 mL/min and the injection volume was 5 μL. The gradient for chromatographic analysis was as follows: 0–1 min, 2% B; 1–9 min, 2–98% B; 9–12 min, 98% B; 12–12.1 min, 98% B; 12.1–15 min, 2% B.

A Q Exactive HF mass spectrometer was used for MS data acquisition. Typical ion source parameters were: sheath gas flow rate: 40; aux gas flow rate: 10; spray voltage in positive ion mode: 3.80 V; spray voltage in negative ion mode: 3.20 V; capillary temperature: 320 °C; aux gas heater temp: 350 °C. To provide more reliable experimental results, the samples were randomly sorted to reduce systematic errors.

### 4.4. Data Processing and Statistical Analysis

For widely targeted metabolomics, Sciex analyst work station software Version 1.6.3 (Sciex, Framingham, MA, USA) was employed for MRM data acquisition and processing. MS raw data files were converted to the TXT format using MSconventer. R software Version 4.0.0 (https://www.r-project.org/ (accessed on 19 September 2020)) was used for peak detection and annotation. For untargeted metabolomics, the acquired mass spectral raw data were imported into Compound Discoverer 3.1 (Thermo Scientific, Wilmington, NC, USA) for data processing, mainly including peak extraction, retention time correction, adduct ion merging, missing value filling, and background peak labeling. Finally, the compound molecular weight, retention time, peak area, and identification results were exported. PCA, PLS-DA, and 200 permutation tests were performed using SIMCA software Version 14.1 (Umetrics AB, Umeå, Sweden). The metabolic pathway analysis of the differentially expressed metabolites was performed by the online tool MetaboAnalyst (https://www.metaboanalyst.ca/ (accessed on 11 March 2022)).

### 4.5. Metabolite Identification and Analysis of Differentially Expressed Metabolites

Metabolites were identified by combining the mzCloud (https://www.mzcloud.org/ (accessed on 1 March 2022)), ChemSpider (https://www.chemspider.com/(accessed on 1 March 2022)), HMDB (https://hmdb.ca/ (accessed on 1 March 2022)), KEGG (https://www.genome.jp/kegg/compound/ (accessed on 1 March 2022)), and LipiMaps (https://www.lipidmaps.org/ (accessed on 1 March 2022)) databases. The main parameters for metabolite identification are precursor mass tolerance, < 5 ppm; fragment mass tolerance, < 10 ppm, RT tolerance, < 0.2 min. The differentially expressed metabolites were identified based on variable importance in a project (VIP) > 1 in the PLS-DA model and the Student’s t test (*p* < 0.05) using SPSS software Version 19.0 (IMB Corp., Armonk, NY, USA). Differentially expressed metabolite classification pie charts and HCA charts were drawn by SPSS software Version 19.0. The relative abundance of the top 20 metabolites with VIP was log-normalized, the 1–20 metabolites correspond to Appendix A, and the relative abundance charts were drawn with GraphPad Prism software Version 9.3 (Graphpad, San Diego, CA, USA).

## Figures and Tables

**Figure 1 molecules-27-03645-f001:**
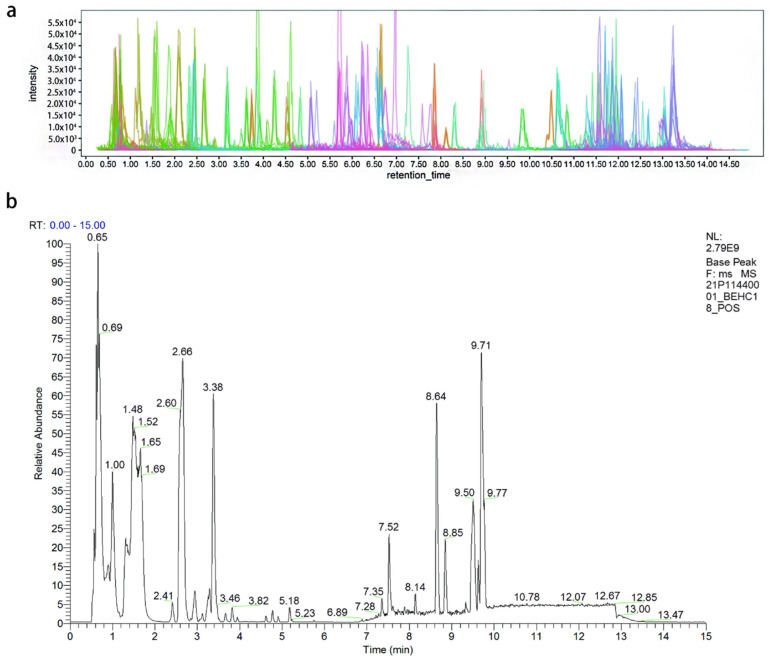
(**a**) Representative total ion chromatogram by the widely targeted metabolomics approach. (**b**) Base peak chromatogram by the untargeted metabolomics approach.

**Figure 2 molecules-27-03645-f002:**
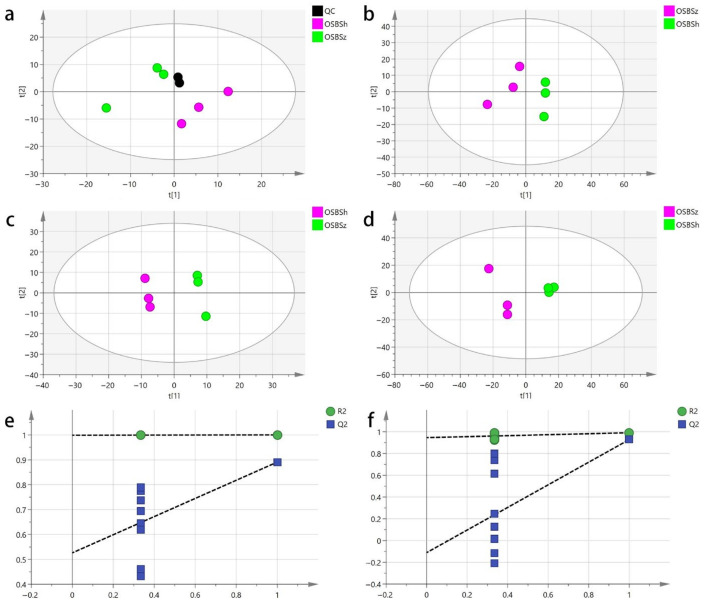
(**a**,**b**) PCA model score of the widely targeted metabolomics approach (**a**) and the untargeted metabolomics approach (**b**). (**c**,**d**) PLS-DA model score of the widely targeted metabolomics approach (**c**) and the untargeted metabolomics approach (**d**). (**e**,**f**) Statistical validation of the PLS-DA model using permutation analysis for the widely targeted metabolomics approach (**e**) and the untargeted metabolomics approach (**f**).

**Figure 3 molecules-27-03645-f003:**
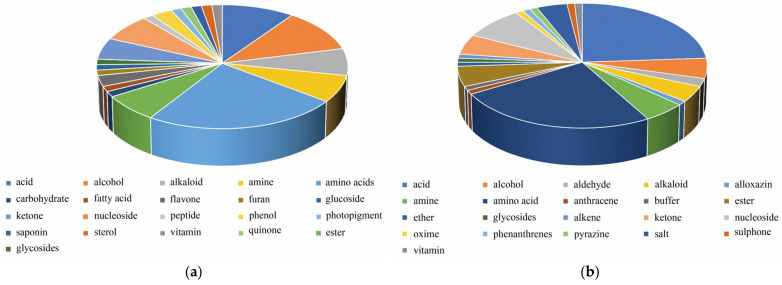
Differentially expressed metabolite classification pie chart. (**a**) Widely targeted metabolomics. (**b**) Untargeted metabolomics.

**Figure 4 molecules-27-03645-f004:**
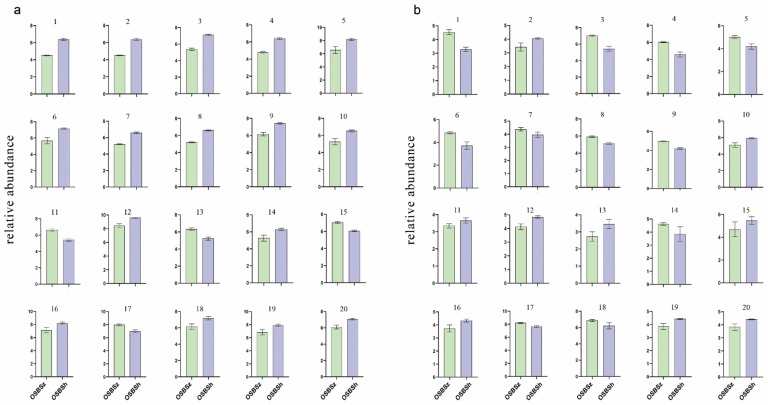
The top 20 differentially expressed metabolites. (**a**) Widely targeted metabolomics. (**b**) Untargeted metabolomics.

**Figure 5 molecules-27-03645-f005:**
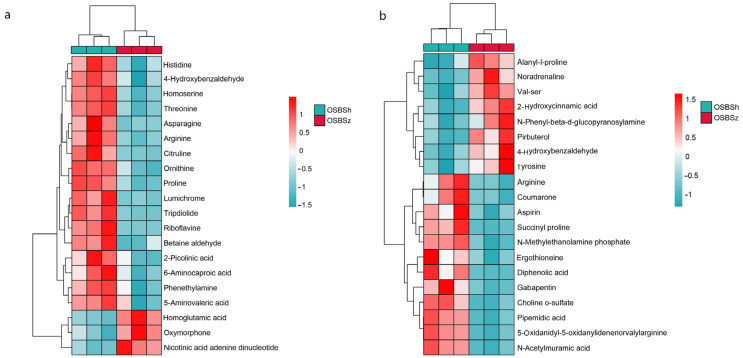
Hierarchical cluster analysis for OSBSz and OSBSh. (**a**) Widely targeted metabolomics. (**b**) Untargeted metabolomics.

**Figure 6 molecules-27-03645-f006:**
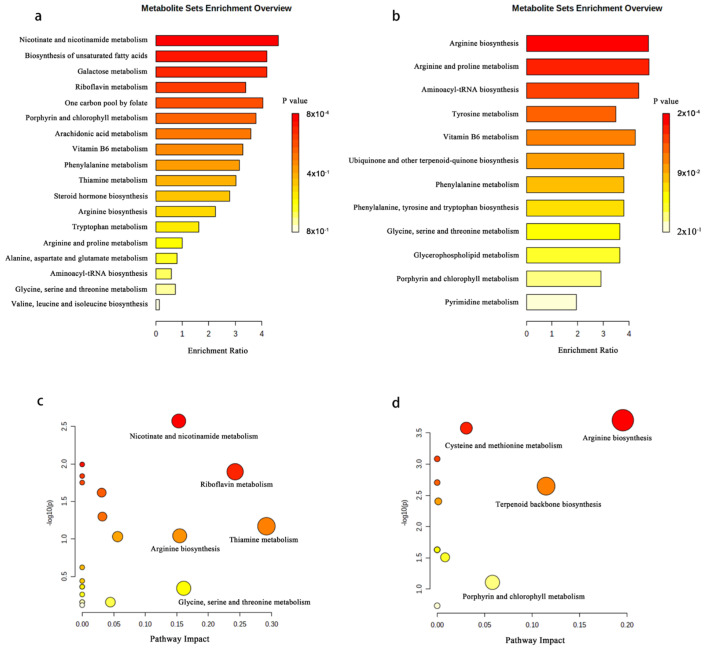
(**a**,**b**) Differentially expressed metabolic pathway maps of OSBSz and OSBSh. (**a**) Widely targeted metabolomics. (**b**) Untargeted metabolomics. (**c**,**d**) Differentially expressed metabolite enrichment analysis of OSBSz and OSBSh. (**c**) Widely targeted metabolomics. (**d**) Untargeted metabolomics.

## Data Availability

Not applicable.

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
