# Peer review of "Comparison of Widely Targeted Metabolomics and Untargeted Metabolomics of Wild Ophiocordyceps sinensis"

_molecules, 2022, doi:10.3390/molecules27113645_

Round 1
Reviewer 1 Report
I don't think there is enough novelty/significance in this work and the English fluency is very poor and also enough references were not cited. Literature search by the author about this work is very poor. I reject this manuscript for publication.
Author Response
Reply to Reviewer #1
Dear Reviewer,
Thank you very much for your time involved in reviewing the manuscript and your very encouraging comments on the merits.
Comments: “I don't think there is enough novelty/significance in this work and the English fluency is very poor and also enough references were not cited. Literature search by the author about this work is very poor. I reject this manuscript for publication.”
Response:We appreciate your clear and detailed feedback and hope that the explanation has fully addressed all of your concerns. Your comments made us aware of the problem with the article. Therefore, fully consider your comments and make revisions in the process of rewriting the article. First of all, this paper was a comparative study on the metabolome of Ophiocordyceps sinensis, hoping that the comparison results of the two methods can provide methodological guidance for Ophiocordyceps sinensis. Secondly, the rewritten article has been revised by professional teachers and language editing companies, hoping to ensure your smooth reading. We also strengthened reference reading and citation, with 43 references cited in our study. Your comments are very important to the revision of the article, and we look forward to your comments.
We attach the modified documents and records.

Author Response
Reply to Reviewer #2
Dear Reviewer,
Thank you very much for giving us an opportunity to revise our manuscript, and we also appreciate you very much for your positive and constructive comments and suggestions on our manuscript.
Comments: “This manuscript describes the explore metabolites present in stroma and sclerotia of Ophiocordyceps sinensis. Researchers have used both targeted and untargeted approaches to address the following:
- The difference between the metabolites between the stroma and sclerotia.
- Comparing the differences by using various basic Stastical analysis.
- Comparing general metabolite profile towards biosynthesis and pathway analysis.
- Comparing targeted Vs untargeted metabolomics approaches.
Here experimental design seems to be quite simple for both the omics analysis, but the conclusions and the data interpretation done by the authors are of highly importance that demonstrates advantages between targeted and untargeted methods. Here they have just done the profiling experiment and the data has been reported as just identified species without proper conclusions or interpretations on the identified metabolites.”
Thank you very much for your opinion. We have rewritten the article based on your comments, placing more emphasis on the methods, results, and initial discussion of the mechanism. In the remainder of this letter, we discuss each of your comments individually along with our corresponding responses. To facilitate this discussion, we first retype your comments in italic font and then present our responses to the comments.
Comment 1: Results/methods section should contain the details of obtaining the 778 metabolites, where did they get these transitions and how did they identify the metabolites.
Response 1: Thank you very much for your professional review, which was a great help in revising the article. The identification of metabolites is the key and difficult point of the metabolome. We have highlighted this work in the article, and the methods and results are presented in the article. Specifically, databases for metabolite identification were described in the “Methods”.
Comment 2,3: Results section needs to be re written, this section reads to me like methods section. Similar sentences are repeated in both of these sections. In the results section, it would be better to explain the obtained results in detail and with detailed explanations of its importance.
Response 2,3: Your comment reminds us to focus on the description of the results, which is crucial to the structure of the article. We have rewritten the Results section to describe the research results in detail. On the other hand, we also explain the obtained results in detail and with detailed explanations of its importance.
Comment 4: The detailed list of the both the targeted metabolites and untargeted metabolites masses need to be disclosed in the supplementary list. This will help the readers to replicate the methodology.
Response 4: Thank you for your careful observation to make the revision of the article more specific. We have added details of differential metabolites in the article, as shown in Table 1 and Table 2.
Table 1 Differential metabolites in samples of the widely targeted metabolomics
|
Number |
Name |
m/z |
VIP |
P-vlue |
RT(min) |
|
1 |
N-Acetyl-L-phenylalanine |
206.10 |
3.910 |
0.0058 |
5.877 |
|
2 |
5-Methyltetrahydrofolic acid |
458.20 |
3.479 |
0.0006 |
3.280 |
|
3 |
Riboflavine |
377.10 |
2.906 |
0.0064 |
5.056 |
|
4 |
Tripdiolide |
377.20 |
2.787 |
0.0110 |
5.136 |
|
5 |
N-Acetyl-L-leucine |
174.10 |
2.721 |
0.0030 |
5.377 |
|
6 |
4-Hydroxy-3-methoxymandelate |
181.10 |
2.431 |
0.0015 |
3.273 |
|
7 |
Bromocriptine |
654.20 |
2.346 |
0.0351 |
12.035 |
|
8 |
Lumichrome |
243.10 |
2.051 |
0.0034 |
6.697 |
|
9 |
N1-Methyl-2-pyridone-5-carboxamide |
153.10 |
2.049 |
0.0033 |
2.628 |
|
10 |
Nicotinic acid adenine dinucleotide |
664.10 |
2.023 |
0.0006 |
0.753 |
|
11 |
Tetrahymanol |
443.40 |
1.987 |
0.0130 |
10.573 |
|
12 |
Citrostadienol |
427.40 |
1.906 |
0.0025 |
12.285 |
|
13 |
11β-Hydroxyandrost-4-ene-3,17-dione |
303.20 |
1.851 |
0.0231 |
6.838 |
|
14 |
Pyridoxine |
170.10 |
1.811 |
0.0417 |
1.552 |
|
15 |
Trans-zeatin-riboside |
352.00 |
1.717 |
0.0439 |
4.310 |
|
16 |
3-(Carboxymethylamino)propanoic acid |
148.10 |
1.712 |
0.0276 |
0.672 |
|
17 |
L-Arginine |
175.10 |
1.704 |
0.0022 |
0.600 |
|
18 |
2-Picolinic acid |
124.00 |
1.701 |
0.0236 |
1.326 |
|
19 |
Melibiose |
341.10 |
1.699 |
0.0011 |
4.546 |
|
20 |
Salsolinol |
180.10 |
1.695 |
0.0113 |
2.266 |
|
21 |
L-Homoserine |
120.10 |
1.680 |
0.0005 |
0.671 |
|
22 |
L-Threonine |
120.10 |
1.673 |
0.0004 |
0.640 |
|
23 |
Gemcitabine |
264.10 |
1.671 |
0.0290 |
0.745 |
|
24 |
L-Homoglutamic acid |
162.10 |
1.665 |
0.0176 |
0.728 |
|
25 |
Hesperidin |
611.20 |
1.661 |
0.0002 |
6.809 |
|
26 |
Flavin adenine dinucleotide (FAD) |
786.20 |
1.658 |
0.0386 |
4.082 |
|
27 |
Isodiospyrin |
375.10 |
1.648 |
0.0332 |
11.278 |
|
28 |
3-Aminoisobutanoic acid |
104.10 |
1.620 |
0.0154 |
0.940 |
|
29 |
Cupressuflavone |
539.10 |
1.608 |
0.0207 |
9.667 |
|
30 |
Aminomalonic acid |
120.00 |
1.601 |
0.0004 |
0.655 |
|
31 |
L-Citrulline |
176.10 |
1.599 |
0.0075 |
0.660 |
|
32 |
N,N-Dihydroxy-L-phenylalanine |
198.10 |
1.595 |
0.0235 |
2.315 |
|
33 |
Docosahexaenoic acid |
329.20 |
1.572 |
0.0204 |
12.603 |
|
34 |
5-Methyl-2-furaldehyde |
111.00 |
1.566 |
0.0144 |
0.755 |
|
35 |
13(S)-HPOT |
311.20 |
1.538 |
0.0104 |
11.465 |
|
36 |
5-Carboxyvanillic acid |
213.00 |
1.522 |
0.0355 |
3.522 |
|
37 |
Isopulegol |
155.10 |
1.483 |
0.0001 |
2.460 |
|
38 |
Serotonin |
177.10 |
1.467 |
0.0280 |
2.612 |
|
39 |
Dipterocarpol |
443.40 |
1.466 |
0.0038 |
12.052 |
|
40 |
1-Aminocyclopropanecarboxylic acid |
102.00 |
1.460 |
0.0020 |
0.700 |
|
41 |
Betaine aldehyde |
102.10 |
1.443 |
0.0495 |
0.753 |
|
42 |
Phenethylamine |
122.10 |
1.441 |
0.0222 |
2.820 |
|
43 |
Thiamine |
265.10 |
1.425 |
0.0386 |
0.701 |
|
44 |
7-Hydroxyflavone |
239.10 |
1.418 |
0.0421 |
0.562 |
|
45 |
Cytosine |
112.10 |
1.412 |
0.0285 |
0.630 |
|
46 |
6,8-Diprenylnaringenin |
409.20 |
1.409 |
0.0017 |
12.621 |
|
47 |
D-Proline |
116.10 |
1.366 |
0.0438 |
0.755 |
|
48 |
L-Ornithine |
133.10 |
1.365 |
0.0129 |
0.550 |
|
49 |
5-(3-pyridyl)-2-hydroxytetrahydrofuran |
166.10 |
1.357 |
0.0111 |
2.803 |
|
50 |
2-carboxybenzaldehyde |
151.00 |
1.333 |
0.0374 |
3.072 |
|
51 |
L-Proline |
116.10 |
1.309 |
0.0166 |
0.730 |
|
52 |
4-Hydroxybenzaldehyde |
123.00 |
1.287 |
0.0313 |
1.905 |
|
53 |
6-Aminocaproic acid |
132.10 |
1.282 |
0.0170 |
1.837 |
|
54 |
Sarsasapogenin |
417.30 |
1.276 |
0.0167 |
12.084 |
|
55 |
L-Histidine |
156.10 |
1.269 |
0.0245 |
0.590 |
|
56 |
Vanillic acid |
169.00 |
1.251 |
0.0460 |
3.776 |
|
57 |
(+)-Pteryxin |
387.10 |
1.236 |
0.0343 |
11.106 |
|
58 |
L-Asparagine |
133.10 |
1.223 |
0.0400 |
0.640 |
|
59 |
p-Aminobenzoate |
138.10 |
1.220 |
0.0273 |
0.736 |
|
60 |
Carnosol |
329.20 |
1.202 |
0.0026 |
12.147 |
|
61 |
5-Aminolevulinate |
132.10 |
1.201 |
0.0230 |
1.555 |
|
62 |
(S)-(-)-perillyl alcohol |
135.10 |
1.200 |
0.0450 |
11.602 |
|
63 |
Ethyl 3,4,5-trimethoxybenzoate |
241.10 |
1.170 |
0.0047 |
0.529 |
|
64 |
5-Aminovaleric acid |
118.10 |
1.157 |
0.0452 |
1.108 |
|
65 |
Cardanol |
303.30 |
1.148 |
0.0293 |
12.688 |
|
66 |
1H-Indole-2,3-dione |
148.00 |
1.145 |
0.0467 |
2.794 |
|
67 |
Oxymorphone |
302.10 |
1.133 |
0.0405 |
11.184 |
|
68 |
Ailanthone |
377.20 |
1.121 |
0.0307 |
3.687 |
|
69 |
β-sitosterol |
397.40 |
1.085 |
0.0096 |
12.930 |
|
70 |
5,6-DHET |
339.30 |
1.080 |
0.0264 |
13.034 |
|
71 |
Anacrotine |
352.20 |
1.024 |
0.0233 |
9.550 |
Table 2 Differential metabolites in samples of the untargeted metabolomics
|
Number |
Name |
m/z |
VIP |
P-vlue |
RT(min) |
|
1 |
Sancycline |
414.14 |
2.349 |
0.0190 |
5.185 |
|
2 |
Vignatic acid B |
519.29 |
2.284 |
0.0151 |
4.656 |
|
3 |
Glycyrin |
382.14 |
2.191 |
0.0123 |
7.114 |
|
4 |
4'-O-Methylkanzonol W |
350.12 |
2.134 |
0.0182 |
7.115 |
|
5 |
Pirbuterol |
240.15 |
2.009 |
0.0362 |
0.713 |
|
6 |
Neotame |
378.22 |
1.978 |
0.0126 |
4.295 |
|
7 |
Rubrophen |
380.13 |
1.963 |
0.0124 |
7.810 |
|
8 |
5'-Dehydroadenosine |
265.08 |
1.916 |
0.0058 |
3.025 |
|
9 |
Nï‰-hydroxy-L-arginine |
190.11 |
1.846 |
0.0102 |
0.607 |
|
10 |
Geniposidic acid |
374.12 |
1.840 |
0.0031 |
4.467 |
|
11 |
3-Phenylpropanoic acid |
150.07 |
1.797 |
0.0457 |
4.055 |
|
12 |
L-Arginine |
174.11 |
1.783 |
0.0210 |
0.625 |
|
13 |
Phthalic acid |
166.03 |
1.716 |
0.0327 |
0.894 |
|
14 |
Methionylleucine |
262.14 |
1.652 |
0.0132 |
3.654 |
|
15 |
Epalrestat |
319.03 |
1.635 |
0.0157 |
0.613 |
|
16 |
N-Phenyl-beta-D-glucopyranosylamine |
255.11 |
1.632 |
0.0160 |
2.719 |
|
17 |
Tes (buffer) |
229.06 |
1.618 |
0.0427 |
0.648 |
|
18 |
4-Pyridoxic acid |
183.05 |
1.592 |
0.0288 |
2.897 |
|
19 |
(2R)-2-î²-gdimboa |
373.10 |
1.584 |
0.0148 |
1.071 |
|
20 |
Ethosuximide |
141.08 |
1.571 |
0.0029 |
2.716 |
|
21 |
12-Azabenz[a]anthracene |
229.09 |
1.569 |
0.0064 |
9.210 |
|
22 |
Vorinostat |
264.15 |
1.551 |
0.0264 |
3.953 |
|
23 |
(Methylthio)pyrazine |
126.03 |
1.540 |
0.0055 |
0.695 |
|
24 |
Nicotinamide adenine dinucleotide |
663.11 |
1.537 |
0.0189 |
0.979 |
|
25 |
Acmimycin |
346.17 |
1.512 |
0.0052 |
1.228 |
|
26 |
Pentothal |
242.11 |
1.439 |
0.0071 |
2.868 |
|
27 |
Phenylethyl alcohol |
122.07 |
1.438 |
0.0016 |
4.042 |
|
28 |
Coumarin |
146.04 |
1.433 |
0.0033 |
1.208 |
|
29 |
Pentobarbital |
226.13 |
1.425 |
0.0156 |
0.710 |
|
30 |
Doxercalciferol |
412.33 |
1.417 |
0.0089 |
9.951 |
|
31 |
Vitamin D2 |
396.34 |
1.416 |
0.0418 |
9.957 |
|
32 |
2-Hydroxycinnamic acid |
164.05 |
1.409 |
0.0019 |
1.206 |
|
33 |
5,10-Methenyltetrahydrofolate |
456.16 |
1.409 |
0.0010 |
0.697 |
|
34 |
Bicine |
163.08 |
1.407 |
0.0254 |
0.655 |
|
35 |
Delta-guanidinovaleric acid |
159.10 |
1.399 |
0.0021 |
1.085 |
|
36 |
L-Alanyl-L-proline |
186.10 |
1.376 |
0.0006 |
0.706 |
|
37 |
Noradrenaline |
169.07 |
1.374 |
0.0006 |
0.706 |
|
38 |
5-Oxidanidyl-5-oxidanylidenenorvalylarginine |
302.15 |
1.372 |
0.0086 |
0.704 |
|
39 |
Febrifugine |
301.14 |
1.354 |
0.0369 |
4.430 |
|
40 |
L-Tyrosine |
181.07 |
1.353 |
0.0006 |
1.040 |
|
41 |
Hydroxytorsemide |
364.12 |
1.338 |
0.0008 |
0.665 |
|
42 |
L-dopa |
197.07 |
1.324 |
0.0220 |
0.992 |
|
43 |
Aspirin |
180.04 |
1.322 |
0.0213 |
0.991 |
|
44 |
Piracetam |
142.07 |
1.318 |
0.0225 |
1.405 |
|
45 |
Boldenone undecylenate |
452.33 |
1.310 |
0.0129 |
9.950 |
|
46 |
γ-L-glutamyl-L-tyrosine |
310.12 |
1.303 |
0.0027 |
2.800 |
|
47 |
N-acetyl-L-tyrosine |
223.08 |
1.297 |
0.0248 |
3.208 |
|
48 |
Val-ser |
204.11 |
1.286 |
0.0006 |
0.707 |
|
49 |
Methohexital |
262.13 |
1.284 |
0.0147 |
3.621 |
|
50 |
Succinyl proline |
215.08 |
1.278 |
0.0174 |
0.991 |
|
51 |
Benzoic acid |
122.04 |
1.275 |
0.0028 |
0.744 |
|
52 |
Piperonyl sulfoxide |
324.18 |
1.272 |
0.0163 |
10.098 |
|
53 |
Reticuline |
329.16 |
1.270 |
0.0141 |
0.652 |
|
54 |
Liafensine |
366.19 |
1.258 |
0.0013 |
9.830 |
|
55 |
Dihydroxyphenylalanine |
197.07 |
1.245 |
0.0202 |
0.884 |
|
56 |
Lumichrome |
242.08 |
1.244 |
0.0138 |
5.674 |
|
57 |
D-Xylonic acid |
166.05 |
1.243 |
0.0044 |
0.649 |
|
58 |
DL-Mevalonic acid |
148.07 |
1.233 |
0.0073 |
0.693 |
|
59 |
Ethenodeoxyadenosine |
275.10 |
1.231 |
0.0411 |
1.428 |
|
60 |
Pro-gln |
243.12 |
1.224 |
0.0034 |
0.692 |
|
61 |
Mebutamate |
232.14 |
1.222 |
0.0116 |
0.718 |
|
62 |
Pipemidic acid |
303.13 |
1.221 |
0.0068 |
1.349 |
|
63 |
γ-Glutamyl-S-(1-propenyl) cysteine sulfoxide |
306.09 |
1.221 |
0.0451 |
3.240 |
|
64 |
11-Nitro-1-undecene |
199.16 |
1.209 |
0.0355 |
6.186 |
|
65 |
Asp-tyr |
296.10 |
1.203 |
0.0015 |
2.456 |
|
66 |
7-Formyldehydrothalicsimidine |
411.17 |
1.199 |
0.0163 |
0.680 |
|
67 |
N2-(1-Carboxyethyl)-2'-deoxyguanosine |
339.12 |
1.191 |
0.0184 |
1.391 |
|
68 |
Deoxyadenosine |
251.10 |
1.191 |
0.0005 |
3.118 |
|
69 |
Coumarone |
118.04 |
1.190 |
0.0272 |
1.210 |
|
70 |
6-Hydroxy-5-methoxyindole glucuronide |
339.10 |
1.172 |
0.0290 |
4.627 |
|
71 |
1-Linoleyl-sn-glycerol 3-phosphate |
434.24 |
1.170 |
0.0394 |
9.579 |
|
72 |
Choline o-sulfate |
183.06 |
1.160 |
0.0058 |
0.664 |
|
73 |
4-Hydroxybenzaldehyde |
122.04 |
1.157 |
0.0381 |
1.216 |
|
74 |
L-Egothioneine |
229.09 |
1.154 |
0.0274 |
0.699 |
|
75 |
1-Stearyl estercitric acid |
444.31 |
1.148 |
0.0036 |
9.412 |
|
76 |
NS-102 |
261.07 |
1.144 |
0.0014 |
4.748 |
|
77 |
N-Ribosylhistidine |
287.11 |
1.141 |
0.0038 |
0.660 |
|
78 |
N-Acetylmuramic acid |
293.11 |
1.134 |
0.0177 |
1.363 |
|
79 |
4-Aminopyridine |
94.05 |
1.134 |
0.0206 |
0.627 |
|
80 |
Porphobilinogen |
226.10 |
1.117 |
0.0209 |
2.900 |
|
81 |
N-Phenylacetylglutamic acid |
265.09 |
1.108 |
0.0029 |
2.371 |
|
82 |
Carglumic acid |
190.06 |
1.100 |
0.0143 |
0.605 |
|
83 |
N-Methylethanolamine phosphate |
155.04 |
1.093 |
0.0420 |
0.654 |
|
84 |
N-Acetyl-L-glutamic acid |
189.06 |
1.091 |
0.0172 |
1.183 |
|
85 |
N2-(carboxymethyl)arginine |
232.12 |
1.089 |
0.0172 |
0.690 |
|
86 |
Lodenosine |
253.10 |
1.084 |
0.0351 |
9.214 |
|
87 |
Arecoline |
155.09 |
1.070 |
0.0291 |
0.715 |
|
88 |
Diphenolic acid |
286.12 |
1.067 |
0.0079 |
0.659 |
|
89 |
Gabapentin |
171.13 |
1.052 |
0.0414 |
4.803 |
|
90 |
γ-Aminobutyric acid |
103.06 |
1.052 |
0.0111 |
0.639 |
|
91 |
Phe-gln |
293.14 |
1.043 |
0.0090 |
3.321 |
|
92 |
Bardoxolone methyl |
505.32 |
1.029 |
0.0210 |
9.358 |
|
93 |
Indospicine |
173.12 |
1.017 |
0.0464 |
1.433 |
|
94 |
3'-Deaminofusarochromanone |
277.13 |
1.015 |
0.0431 |
2.437 |
|
95 |
2'-Deoxycytidine |
227.09 |
1.014 |
0.0314 |
4.673 |
|
96 |
Troxipide |
294.16 |
1.003 |
0.0140 |
2.704 |
Comment 5: It is not clear from the figures and the methods section, that how many replicates did they use to obtain the conclusions, from the PCA plots it seems to be 3 replicates, are they biological or technical replicates.
Response 5: We are very sorry for our negligence of describe. In fact, we used 3 biological replicates, and added to the Methods section of the manuscript.
Comment 6: The figures in the article is not clear, it needs to be redrawn by increasing the font.
Response 6: It is really true as Reviewer suggested that figures in the article is not clear. Therefore, we have made appropriate adjustments to the image.
Comment 7: Figure 2, column chart needs to be redrawn in another format, the stacked format is bit confusing and can’t read any text in the figure.
Response 7: We grateful your advice, as you said that the stacked format is bit confusing and can’t read any text in the figure. We redrawn the column chart and plotted the relative abundance of column chart the top 20 metabolites according to the VIP value to observe the difference between OSBSz and OSBSh.
Comment 8: There is a lack of explanation of the observed metabolites in respective stroma and sclerotia samples. I suggest researchers to explain the reasons behind these metabolites and why do we observe the similar and different in this stroma and sclerotia samples.
Response 8: Those comments are all valuable and very helpful for revising and improving our paper, as well as the important guiding significance to our researches. In the Results and Discussion section of the article, we have added descriptions of some metabolites. We found important metabolites responsible for the difference between OSBSz and OSBSh, and these metabolites play important roles in the growth of the stroma and sclerotia.
Comment 9: I would suggest to researchers to describe the mechanisms behind the common components that were found in these two methods.
Response 9: Special thanks to you for your good comments. Based on your suggestion we also explored he common components that were found in these two methods in the Discussion section and they have a particular role for OSBSz and OSBSh.
Comment 10: The article needs to rewritten focusing the broad profiling experiment. Some of the experimental details are repeated both in the methods section and also in the results section. Repetition of the same thing multiple times is not good.
Response 10: Thank you for taking the time to review our manuscript, as you said, there are many problems with this article and your suggestions are crucial to the revision of the manuscript. Therefore, in the process of rewriting the article, we consider your comments to enrich the article. In the discussion section of the article, there is also more emphasis on discussing the mechanism of difference.
We would like to take this opportunity to thank you for all your time involved and this great opportunity for us to improve the manuscript. We hope you will find this revised version satisfactory.

Reviewer 3 Report
The manuscript described performing both targeted and untargeted approach to profile the metabolome of Ophiocordyceps sinensis. Overall, the study design and result is OK. However, a significant writing quality improvement is required for this manuscript. I just listed some typos or grammar errors caught by me. The authors need to read the manuscript carefully and fix all the typos and grammar errors.
(1) Line 49 delete “analysis techniques”
(2)Line 53, change “different samples. Such as” to “different samples, such as”
(3)Line 65-66 change “to detected” to “to detect”
(4)Line 233 “liquid phase” should be “liquid chromatography”
(5)Line 233, 237 “mass spectrometry” should be “mass spectrometer”
(6)Line 245 insert spaces to “1minat” and “100mg”
(7)Line 259 “drained” should be “dried down”
Other concerns:
(1)Line 113, could authors’ column separate L or D-arginine? If not, the authors should delete L or D before arginine.
(2)Figure 2 and 4, the authors should provide high resolution figure.
Author Response
Reply to Reviewer #3
We are very grateful to your comments for the manuscript. According with your advice, we tried our best to amend the relevant part and made some changes in the manuscript. These changes will not influence the content and framework of the paper. All of your questions were answered below.
Comments: “The manuscript described performing both targeted and untargeted approach to profile the metabolome of Ophiocordyceps sinensis. Overall, the study design and result is OK. However, a significant writing quality improvement is required for this manuscript. I just listed some typos or grammar errors caught by me. The authors need to read the manuscript carefully and fix all the typos and grammar errors.
(1) Line 49 delete “analysis techniques”
(2)Line 53, change “different samples. Such as” to “different samples, such as”
(3)Line 65-66 change “to detected” to “to detect”
(4)Line 233 “liquid phase” should be “liquid chromatography”
(5)Line 233, 237 “mass spectrometry” should be “mass spectrometer”
(6)Line 245 insert spaces to “1minat” and “100mg”
(7)Line 259 “drained” should be “dried down”
Other concerns:
(1)Line 113, could authors’ column separate L or D-arginine? If not, the authors should delete L or D before arginine.
(2)Figure 2 and 4, the authors should provide high resolution figure.”
Comment 1:(1) Line 49 delete “analysis techniques”
(2)Line 53, change “different samples. Such as” to “different samples, such as”
(3)Line 65-66 change “to detected” to “to detect”
(4)Line 233 “liquid phase” should be “liquid chromatography”
(5)Line 233, 237 “mass spectrometry” should be “mass spectrometer”
(6)Line 245 insert spaces to “1minat” and “100mg”
(7) Line 259 “drained” should be “dried down”
Response 1: Thank you for the detailed review. We have carefully and thoroughly proofread the manuscript to correct all the grammar and typos. We rewritten the article according to the review comments. LetPub for its linguistic assistance during the preparation of this manuscript.
Comment 2:Line 113, could authors’ column separate L or D-arginine? If not, the authors should delete L or D before arginine.
Response 2:Thank you for your valuable advice, in fact, there are differences between L-arginine and D-arginine. L-arginine is an essential amino acid for the human body, which cannot be synthesized by the human body and needs to be ingested from food. D-arginine is derived from L- Arginine is used as raw material, and it is obtained by racemization reaction.
Comment 3:Figure 2 and 4, the authors should provide high resolution figure.
Response 3: It is really true as Reviewer suggested that figures in the article is not clear. We have improved the resolution of the figure.
I wish this revision will be acceptable for publication in your journal.
Thank you for your consideration. I am looking forward to hearing from you.

Round 2
Reviewer 1 Report
The manuscript can be published after minor revisions. See below:
The authors have improved the manuscript quite better from the previous version. I still would suggest the authors to revise the manuscript more thoroughly, especially the abstract part (writing needs to be improved), and make sure to revise all other sections including the conclusion part once or twice.
Author Response
Reply to Reviewer #1
Dear Reviewer,
Thank you very much for giving us an opportunity to revise our manuscript, and we also appreciate you very much for your positive and constructive comments and suggestions on our manuscript.
Comments: “The manuscript can be published after minor revisions. The authors have improved the manuscript quite better from the previous version. I still would suggest the authors to revise the manuscript more thoroughly, especially the abstract part (writing needs to be improved), and make sure to revise all other sections including the conclusion part once or twice.”
Response:We appreciate your clear and detailed feedback and hope that the explanation has fully addressed all of your concerns. Therefore, we carefully revised the abstract, discussion and other sections. Thank you very much for your professional review and pointing out the direction for our manuscript. We sincerely hope that this manuscript can be published and look forward to your review.We would like to take this opportunity to thank you for all your time involved and this great opportunity for us to improve the manuscript. We hope you will find this revised version satisfactory.

Reviewer 2 Report
Thanks for considering all those coments and taking time to revise the manuscript. the paper reads much better now. is it possible to increase the font of x and y axis of figure 4.
Author Response
Dear Reviewer,
Thank you very much for your time involved in reviewing the manuscript and your very encouraging comments on the merits.
Comments: “Thanks for considering all those coments and taking time to revise the manuscript. the paper reads much better now. is it possible to increase the font of x and y axis of figure 4. ”
Response: Thank you very much for your comments in the round 1 of comments, your comments are very professional and detailed. We made some changes to the abstract and discussion, and also adjusted Figure 4. Thanks again for your suggestion, your comments are very important to the revision of the article, and we look forward to your comments.

Reviewer 3 Report
I'm glad to see the authors revised the manuscript carefully. I think the writing quality of the manuscript is OK now. In the previous round of review, I asked the authors if their LC column was able to separate L- or D- metabolites. The authors didn't understand my question. I know the L- or D- are different metabolites and have different physiological functions, but I’m talking about separation of them on a LC column. It is impossible to separate L- or D- metabolites on C18 column without derivatization. The authors should check the raw data carefully to confirm this. The authors should remove L-/D- in text and heatmaps if this is true.
Author Response
Reply to Reviewer #3
Thank you very much for giving us an opportunity to revise our manuscript, and we also appreciate you very much for your positive and constructive comments and suggestions on our manuscript. Your professional advice is essential for the revision of the article.
Comment: “I'm glad to see the authors revised the manuscript carefully. I think the writing quality of the manuscript is OK now. In the previous round of review, I asked the authors if their LC column was able to separate L- or D- metabolites. The authors didn't understand my question. I know the L- or D- are different metabolites and have different physiological functions, but I’m talking about separation of them on a LC column. It is impossible to separate L- or D- metabolites on C18 column without derivatization. The authors should check the raw data carefully to confirm this. The authors should remove L-/D- in text and heatmaps if this is true.”
Response:We are very sorry for misunderstanding your advice before, we take your advice very seriously, and consulted experts and teachers due to our lack of knowledge. As you said, it is impossible to separate L- or D- metabolites on C18 column without derivatization, and we remove L-/D- in text and heatmaps. In addition, we focused on revising the abstract and discussion sections. Thanks again for your professional review.
